# Harnessing explainable artificial intelligence for patient-to-clinical-trial matching: *A proof-of-concept pilot study using phase I oncology trials*

**Satanu Ghosh[1], Hassan Mohammed Abushukair[2], Arjun Ganesan[3], Chongle Pan[3], Abdul Rafeh Naqash[4]***, **Kun Lu[5]***

1 Department of Computer Science, University of New Hampshire, Durham, New Hampshire, United States of America, 2 Jordan University of Science and Technology, Irbid, Jordan, 3 School of Computer Science, University of Oklahoma Norman Campus, Norman, Oklahoma, United States of America, 4 Medical Oncology/ TSET Phase 1 Program, Stephenson Cancer Center, The University of Oklahoma Health Science Campus, Oklahoma City, Oklahoma, United States of America, 5 School of Library and Information Studies, University of Oklahoma Norman Campus, Norman, Oklahoma, United States of America

☯ These authors contributed equally to this work.
* kunlu@ou.edu (KL); abdulrafeh-naqash@ouhsc.edu (ARN)

**Data Availability Statement:** These data contain potential patient identifiers and genomic information that can't be shared openly in the

## Abstract

This study aims to develop explainable AI methods for matching patients with phase 1 oncology clinical trials using Natural Language Processing (NLP) techniques to address challenges in patient recruitment for improved efficiency in drug development. A prototype system based on modern NLP techniques has been developed to match patient records with phase 1 oncology clinical trial protocols. Four criteria are considered for the matching: cancer type, performance status, genetic mutation, and measurable disease. The system outputs a summary matching score along with explanations of the evidence. The outputs of the AI system were evaluated against the ground truth matching results provided by the domain expert on a dataset of twelve synthesized dummy patient records and six clinical trial protocols. The system achieved a precision of 73.68%, sensitivity/recall of 56%, accuracy of 77.78%, and specificity of 89.36%. Further investigation into the misclassified cases indicated that ambiguity of abbreviation and misunderstanding of context are significant contributors to errors. The system found evidence of no matching for all false positive cases. To the best of our knowledge, no system in the public domain currently deploys an explainable AI-based approach to identify optimal patients for phase 1 oncology trials. This initial attempt to develop an AI system for patients and clinical trial matching in the context of phase 1 oncology trials showed promising results that are set to increase efficiency without sacrificing quality in patient-trial matching.

## Introduction

Phase-1 or early-phase clinical trials are uniquely poised within the clinical trial process as they are the first clinical step in drug development, making them the first chance to identify clinical

journal. Select deidentified data can be potentially shared with readers upon specific requests to the corresponding authors. This is imposed by the Institutional Review Board based on HIPAA. We do not have a data access committee. The Institutional Review Board can be reached at 405-271-2045 or irb@ouhsc.edu.

**Funding:** This study was supported by the internal seed DISC/SCC grant at the University of Oklahoma awarded to CP, ARN, and KL. KL is the recipient of the Open-Access Research Publication Funding from OU Libraries. The funders had no role in study design, data collection and analysis, decision to publish, or preparation of the manuscript.

**Competing interests:** The authors have declared that no competing interests exist.

biomarkers and efficacy signals [1]. Optimizing phase 1 trials by a proper phenotypic and genotypic selection of patients represents a promising avenue to rapidly improve drug development, biomarker identification process, and eventually, patient clinical outcomes [2, 3]. Currently, a significant proportion of clinical trials fail. One of the main reasons for the failure is the suboptimal patient selection and recruitment processes [4, 5]. Patient selection and recruitment, especially for phase 1 trials, require assimilation and analysis of copious amounts of patient clinical/genomic data making the process extremely time, skill, and labor-intensive. There is great potential to apply cognitive computing to streamline the oncology phase 1 drug development process, including on-treatment data capture, and achieve cost, time, and research benefits. Although some early promise has been shown with using artificial intelligence (AI) based approaches to identify appropriate patients for clinical trials [4, 6–8], these efforts have yet to focus on cancer-specific phase 1 trials. Patient selection for oncology trials may have very different criteria that require the development of unique systems to match them. For example, the n2c2 shared task selected 13 criteria for matching patients with diabetes to clinical trials [9]. We found none of the 13 criteria applicable for selecting patients for oncology trials. To the best of our knowledge, no system in the public domain currently deploys an explainable AI-based approach to identify optimal patients for phase 1 oncology trials. There is an unmet need to develop customized systems for patient selection for phase 1 oncology clinical trials.

Automated selection of patients for clinical trials is a relatively new domain that requires much exploration. Some recent studies have explored and streamlined patient recruitment in different domains [10–12]. However, most of these studies are either using large synthetic data [12] or proprietary systems [11] (like Watson for Clinical Trial Matching). There is not an open dataset to evaluate patient-clinical-trial matching in phase 1 oncology trials. Similar to Calaprice-Whitty and Salloum [10] our work draws emphasis on evidence-based matching, and we provide an in-depth description of each step used in the system. Moreover, this study is uniquely poised because we work on real world data (modified real patient data) that focuses on early phase clinical trials for cancer.

Historically, most machine-learning-based decision-making systems can be classified as black-box systems where the prediction is not reasoned or explained [13]. For sensitive domains (like health) black-box systems are not considered trustworthy [14–16] because their inner-workings are not understandable to end-users. However, these black-box systems [17, 18] have been developed for patient selection. These systems cannot have a human-in-the-loop because given some input they only provide an output without any rationale or explanation behind the decision taken. One desideratum of explainable decision-making systems is to reduce the cognitive load of users by providing them with all the information that contributed towards the prediction of the system, particularly in risk-averse domains [16, 19].

Our central hypothesis was that utilizing cognitive computing techniques, such as Natural Language Processing (NLP), would improve the efficiency of patient selection and recruitment for phase 1 trials. We aimed to pilot test and develop an unsupervised prototype NLP system that follows a mixed approach combining tools, regular expressions and expert curated rules to match patients with eligible phase 1 clinical trials based on relevant eligibility criteria specified within the clinical trial protocols. In this initial phase, our objective was to demonstrate the feasibility of optimizing and automating certain aspects of patient recruitment to relevant phase 1 trials. This pilot project developed a prototype system that provides a matching score along with explanations of why the score is assigned to that patient-trial matching pair which could make screening and cross-checking much easier for physicians and research teams who work on early phase clinical trials. The long-term goal is to develop an integrated AI-based user-friendly (with human-in-the-loop capability), and intuitive interface to significantly improve workflow efficiency and generate automated *patient-to-trial* matching for phase 1 clinical trials.

## Methods

### Matching criteria

Our approach evaluates four criteria that are important for patient-clinical trial matching in the context of phase 1 oncology trials: (a) Cancer type: While a majority of phase 1 trials tend to be tumor agnostic during dose escalation, some trials are restricted to certain tumors only, especially in the expansion phase. Most phase 1 trials are geared toward patients who have exhausted standard-of-care therapies in the metastatic setting. Thus, we included metastatic disease as a sub-variable based on distant site involvement since almost all phase 1 protocols exclusively include patients with advanced/metastatic disease. (b) Clinical performance status (PS) graded on the eastern cooperative group (ECOG) PS score from 0 to 4, where 0 refers to "highly functional" and 4 is "bed bound.". Several studies have shown that ECOG PS is a surrogate for patient prognosis, with higher scores associated with inferior survival. ECOG PS is usually restricted to 0–1 and occasionally 0–2 in phase 1 trials that include immunotherapies or novel targeted therapies. (c) Measurable or evaluable disease is important from a trial standpoint as it allows serial measurement assessment of target lesions using the response evaluation criteria in solid tumors (RECIST) criteria to evaluate response to therapy. Usually, a lesion in any organ at least 1 cm or above in the short axis or a lymph node 1.5 cm or above at least in the short axis is considered measurable. Bony lesions are generally not included as measurable but are considered evaluable. (d) Mutation status. Several phase 1 trials employ a precision medicine-based approach, only allowing specific DNA-based pathogenic alterations or RNA-based fusions.

### Data

No existing datasets were found suitable for testing patient-to-trial matching in the context of phase 1 oncology trials. Our study created a dataset with twelve synthesized dummy patient records and six clinical trial protocols. Patient records were generated in a deidentified manner from real-life patient records by modifying patient clinical characteristics for variables of interest to pilot-test our method. The clinical protocols were derived from incorporating inclusion/exclusion criteria for certain phase 1 protocols at the Stephenson Cancer Center directly from the publicly available protocols from the Clinicaltrials.gov website using the NCT number of the trial. The physician on the team reviewed each patient and clinical trial protocol pair (a total of 72 pairs) to generate the ground truth for patient-to-trial matches. Only the physician on the team had access to the actual patient record and the entire phase 1 protocols. The data was provided by the physician to the team on May 26th, 2022. All patient identifiers were removed. This research was conducted in compliance with HIPPA standards after appropriate IRB approval (OUHSC IRB number: 14579). The small sample size is due to the labor-intensive process of generating the dataset.

### System overview

The process of the system includes pre-processing, information extraction, and matching. The following flowchart provides a summary of the process (Fig 1).

## Pre-processing

### Named entity recognition

The patient records and clinical trial protocols are unstructured text and, therefore, quite challenging to infer their match. We created a pre-processing pipeline to reduce the noise from both types of text (Fig 2. Medical documents express the most important information in

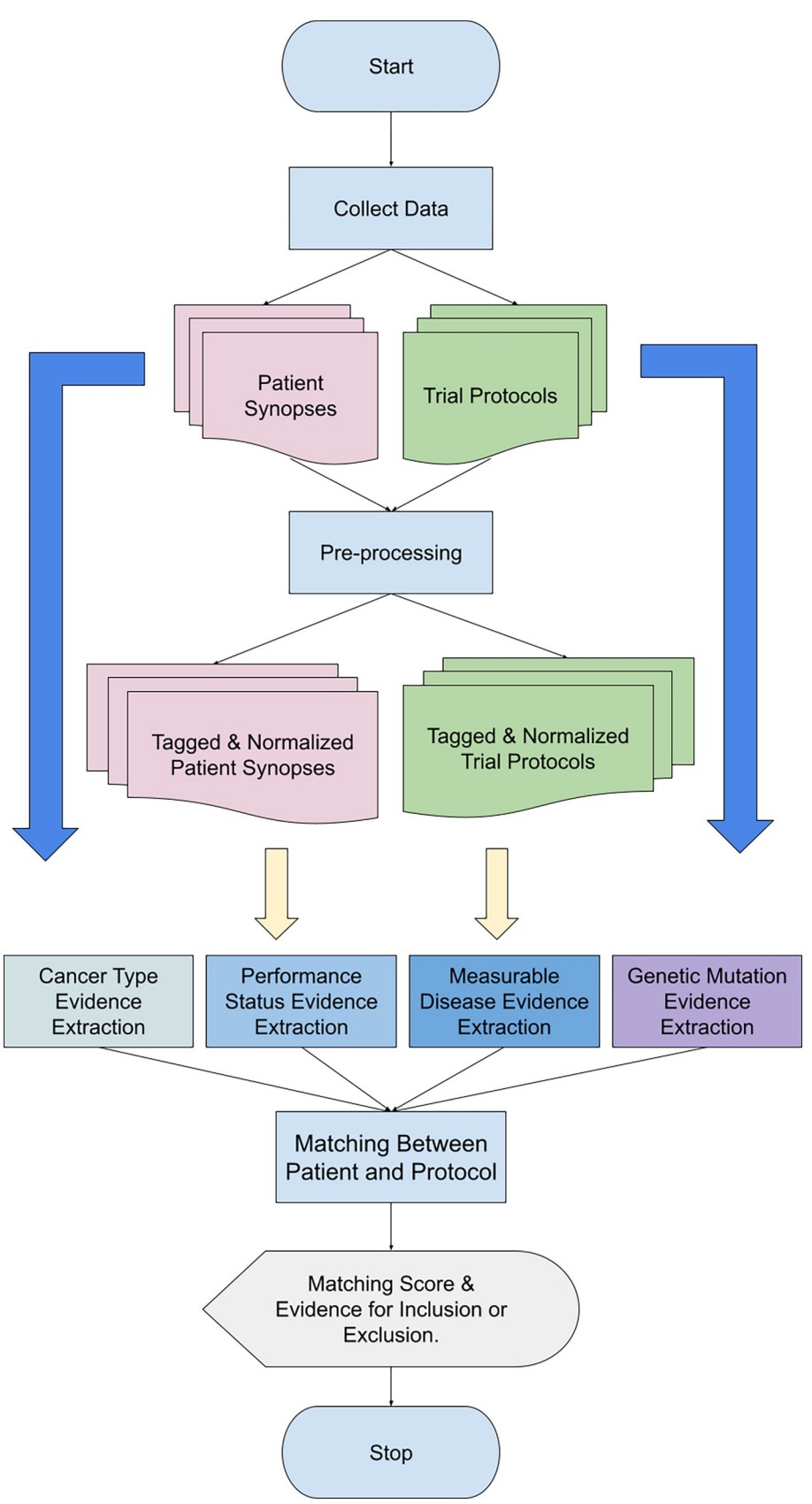

**Fig 1. System flowchart.** The flowchart of our approach is provided in this image. More details about pre-processing, each criterion and matching method are below.

various entities (genes, diseases, drugs, etc.). To get a natural understanding of medical documents, we represented them in terms of entities (both broad and fine-grained categories). We used three tools, SciSpacy [20], DeepPhe [21], and BERN2 [22], to extract all entities from the protocols and patient records. SciSpacy and BERN2 are NLP tools specially built to extract biomedical entities. Both are made using state-of-the-art NLP techniques and have shown excellent performance on benchmark evaluation [20, 22]. DeepPhe is an NLP system that can extract cancer phenotypes from patient records and extract entities as an intermediate step of the process.

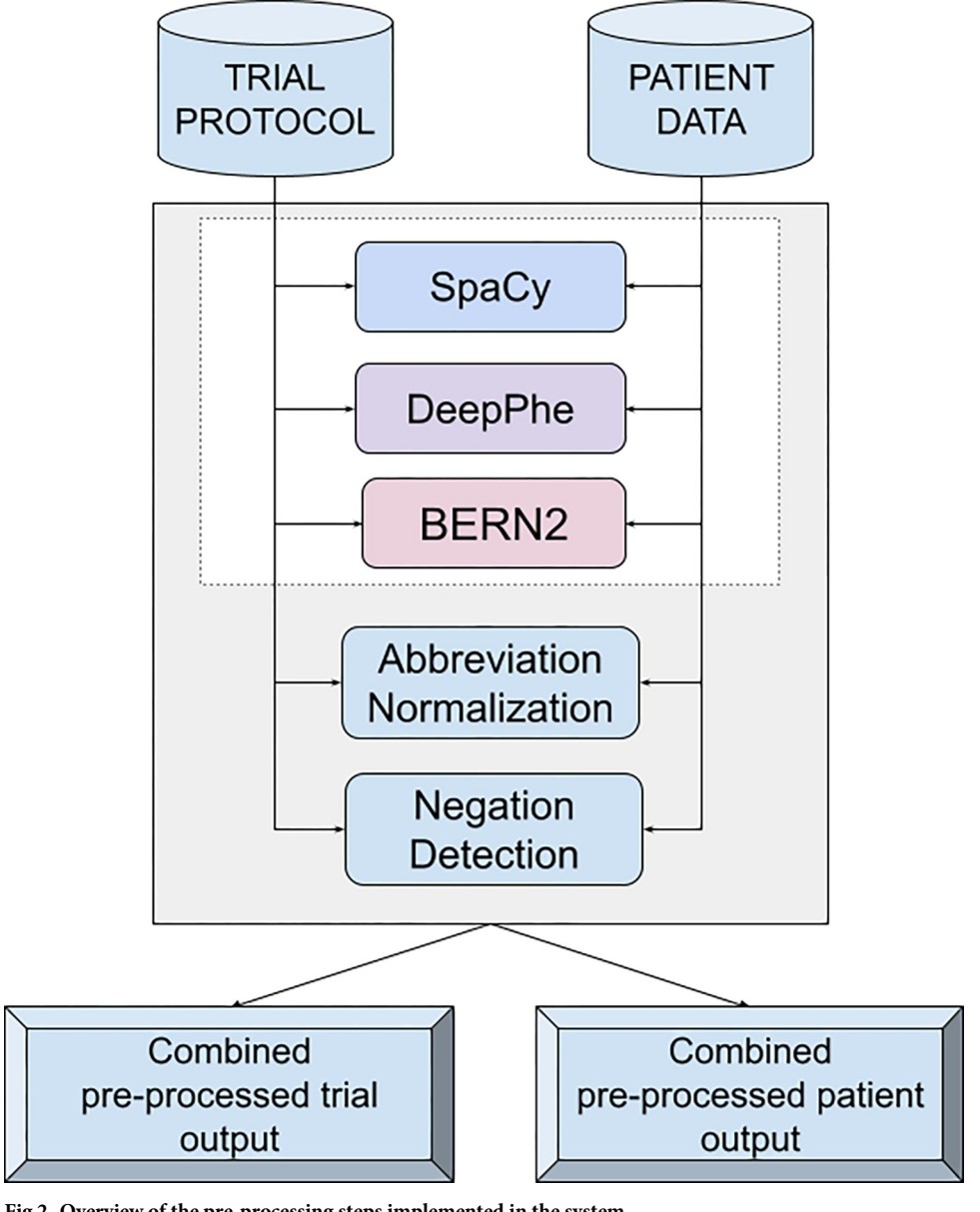

**Fig 2. Overview of the pre-processing steps implemented in the system.**

SciSpacy has several pre-trained models. This work used models trained on BIONLP13CG, JNLPBA, and SciSpacy large biomedical corpus. There are other models of SciSpacy, but we chose these three based on their tagsets and our needs. The output from all three tools was combined based on the offset positions of entities appearing in the text. The output of entity detection from each tagger was sometimes overlapping. For example, we found "mucinous adenocarcinoma" detected by all three taggers but with three different tags, "disease," "ENTITY," and "Adenocarcinoma" by BERN2, SciSpaCy, and DeepPhe, respectively. Our combined output was the entity "mucinous adenocarcinoma," followed by all three tags "disease, ENTITY, Adenocarcinoma," followed by the tagger names "BERN2, SciSpacy, DeepPhe" followed by the character offset position of the token. BERN2 has a different text encoding schema than the other two taggers, and we found that the character offset positions were often different. Therefore, we decided to run a search on the raw text with a window size of 40 to identify the entity detected by BERN2 and rectify its offset values.

## Abbreviation detection and normalization

Acronyms or abbreviations are commonly used in medical documents. They are often misunderstood by the taggers. One convention is to write out the entire term the first time it occurs in the document, and in parentheses, an acronym/abbreviation is given, indicating that the term will be referred to by that acronym/abbreviation throughout the rest of the document. As such, we extracted all such acronyms/abbreviations during preprocessing and mapped them to their full meaning based on their first occurrences using regular expression patterns. The full meaning of every subsequent time the acronym/abbreviation was found in a document was used for matching purposes.

## Negation detection

By default, the Named Entity Recognition (NER) taggers do not understand the negative context in which a particular term is used. For example, if a protocol document mentions that patients are only eligible for a trial if they have not taken a particular medication, the taggers will recognize the medication, but will not understand that the medication is being discussed in a "negative" context. As such, we utilized the NegEx algorithm implemented in Spacy to parse each sentence for specific negating patterns such as "not including" or "except for." For any sentence in which those patterns appear, any token tagged in that sentence is marked as "negated". Such information becomes relevant during the "Patient-Trial" matching step.

Collectively, at the end of the preprocessing step, we found all the relevant entities in the patient records and the protocol data, along with the type of entity, normalized all abbreviations and found if a particular entity was found in a negative context. An example output of the preprocessing step is provided in Fig 3.

## Information extraction and matching

### Cancer type

The cancer type mentioned in patient records and clinical trial protocols was determined by matching the BERN2 tagged "disease" type tokens and DeepPhe tagged tokens to the National Cancer Institute (NCI) thesaurus [23]. More specifically, we used the branch of the concept "Neoplasm" in the NCI Thesaurus (Code C3262) to map the tokens in the text to concepts in the NCI thesaurus. To determine a match, the tokens in patient records and clinical trial protocols are compared to all synonyms and abbreviations of a concept in the NCI thesaurus. Our program normalizes tokens in text and synonyms/abbreviations in thesaurus by converting

| Token | Tag | Start Pos. | End Pos. | Tagger | Acronym meaning | Sentence Position | Is Negated? |
|---|---|---|---|---|---|---|---|
| GGT | Gene, GENE OR GENE PRODUCT, ENTITY | 3592 | 3595 | BERN2, SpaCy, SpaCy | gamma glutamyl transferase | 32 | False |
| uln | ENTITY | 3634 | 3637 | SpaCy | Upper limit of normal | 32 | False |
| liver involvement | disease | 3654 | 3671 | BERN2 | —--- | 32 | True |

**Fig 3. Example of pre-processing output.**

them to lowercase and ordering the words alphabetically to account for expression variations. A two-tiered mapping strategy was adopted first to map tokens to preferred concept names; if that did not result in a match, then synonyms were used. A few general cancer types were excluded because they did not provide useful information to determine a patient-trial match, such as "Malignant Neoplasm," "Solid Neoplasm," and "Metastatic Neoplasm." After this first round of cancer type detection, an effort was made to determine additional patient cancer types by combining the organ mentions (e.g., "lung") in the same sentence as a cancer mention initially detected (e.g., "adenocarcinoma") for a more specific cancer type matching with both cancer site and type (e.g., "lung adenocarcinoma"). This alleviated the issue of expression variations in clinical notes.

In addition to detecting cancer types, we also tried to determine the metastatic status of cancers. We used "Secondary_Malignant" and "Metastasis" tags from DeepPhe as well as token mentions "metastatic" to determine if a cancer mention is metastatic. If metastatic status was detected anywhere in the record, a patient was identified to have metastatic cancer. For a protocol, if the sentence had both a specific cancer mentioned and a metastatic status, this cancer occurrence was considered to be metastatic.

When matching a patient with a protocol in terms of cancer type, we used the mapped concepts in the NCI thesaurus. If the patient's cancer type is a subtype of the cancer type required by a protocol and their metastatic status matches, the patient is considered to have a cancer type matching with the protocol.

### Performance Status (PS)

PS of a patient can be defined as their ability to perform daily tasks. The most common way to assess performance status is using the ECOG scale, which is present in almost all phase 1 oncological trial protocols as an eligibility criterion [24]. However, the ECOG score is not always explicitly mentioned in patient files.

ECOG scores were not tagged by the entity taggers in the preprocessing step; therefore, we used regular expressions to extract the ECOG score. We performed two types of performance status extraction: implicit and explicit from the patient file. In the patient files, sometimes, there were words or brief phrases describing patients' conditions. For example, "active," "doing well," "in good health," and "fatigued," "wheelchair bound," "limited activity," "bed bound" were used to describe good and bad health, respectively. Depending on the words we found, we assigned an ECOG level, i.e., 0–1 for words pertaining to good health and 2–4 for words related to bad health. In explicit extraction, we looked for the keyword "ECOG" in the entire text, and if we found a match, then extracted any integer value present next to it. Protocols always have explicit mentions of the ECOG level. However, we had to extract the sign ($\leq$,

$\geq$, -) next to the "ECOG" keyword to make the proper decisions. After we extracted ECOG levels from both the protocol and the patient files, we checked if the patient satisfied the protocol requirements.

## Measurable disease

One of the most important factors in oncological trials is whether the tumor is measurable. Measurability is only possible if the tumor has a certain size; however, the size may vary depending on the site of the tumor. If the tumor is in bone, it is not measurable; if it is in a solid organ or lymph node, it needs to be greater than 1 cm and 1.5 cm, respectively, to be measurable. So, we needed both the site and size of the tumors. These rules are based on commonly employed response evaluation criteria in solid tumors (RECIST1.1.) criteria in clinical trials [25].

For extracting measurability for patients, we took the output of DeepPhe to detect the site of cancer. We then normalized the site into two categories: solid organs and lymph nodes. Based on the site, we can understand whether it is a solid organ or a lymph node. Next, we extracted the size of the tumor using a regular expression from the patient file. We found that the size information in the patient file can be written in different ways, like (n1 x n2) cm, (n1 x n2 x n3) cm, or (n1.n2) cm. We took the longest dimension in case multiple dimensions were present in tumor size. Then, we took a binary decision of whether the disease is measurable depending on both the site and size. To understand whether a protocol requires measurable disease or not, we check whether a protocol has phrases like "measurable disease" or "measuring". Furthermore, we also consider that the protocol requires a measurable disease only if it does not contain any mention of a "non-measurable disease."

Finally, we selected patients with measurable diseases for protocols with the requirement of measurable diseases. If a protocol did not require measurable disease, then patients with both measurable and non-measurable diseases were considered to qualify for that protocol.

## Genetic mutation

Patients suspected to have cancer are tested for genetic mutations, and the results can be found in their records. Some protocols are specific about the type of mutations that need to be present in the patient to be considered eligible for the protocol. For example, a protocol in our dataset stated that "A known functional mutation (including novel loss of function frameshift or nonsense mutations) in any of the following DDR genes: ARID1A, ATM, ATR, . . ., RAD54L."

Based on the tags that we extracted during the pre-processing step, we narrowed down three tags that were relevant to genetic mutations "gene", "mutation," and "DNA." The tokens corresponding to these tags found in patient files were matched with the tokens corresponding to these tags found in protocol files. If the overlapping tokens were more than zero, then we assumed that the patient was a match for the genetic mutation requirement relevant to the protocol.

In some instances, patients were previously treated with precision medicine-based targeted therapies, and this information is present in their files. Entity taggers are unable to differentiate between gene mutations mentioned in the text and precision medicine-based targeted therapies. However, the context of genetic mutations and gene therapy are different. In our dataset, we found that precision medicine-based gene therapy is always mentioned with other tokens like "anti" as in "anti-HER2 therapy" and "targeted" as in "VEGF-targeted." Therefore, we only identify genetic mutation mentions outside these gene therapy contexts.

## Matching score

Protocols have two distinct sections: Inclusion and Exclusion. We tracked whether our extracted information belonged to the inclusion or exclusion section. The output contained a detailed explanation about the categories (out of four) the patient was a match for within the inclusion criteria and the categories that the patient was a match for within the exclusion criteria of the protocol. The system also calculated a matching score as follows:

$$Score = number\ of\ criteria\ met\ by\ a\ patient / number\ of\ criteria\ required\ by\ a\ protocol \quad (1)$$

So, if a protocol required all four criteria and a patient met two of them, the patient was calculated to have a 0.5 matching score.

# Results

## System output

Our system outputs evidence for each of the four criteria, with *evidence* or *no evidence* outcomes. In addition, a summary of the overall matching score is reported (Fig 4). To evaluate the quality of the AI-generated results, our model was set to consider a matching score greater than 0.5 (meaning there is greater evidence of matching than non-matching) as a matching decision. A higher or lower threshold could be set for more or less stringent matching. However, the most important feature of our system is that it provides all evidence of matching or non-matching to assist human decisions.

## System performance metrics

Our twelve synthesized dummy patient records and six clinical trial protocols resulted in 72 (12*6) case scenarios for match assessment. To assess the performance, we compared system-produced *match/no-match* decisions with ground truth matching decisions provided by our domain experts. Of those, 19 (26.39%) scenarios were considered as a match by our model of which five were false positives, and 53 (73.61%) were non-matching with eleven false negatives. Table 1 demonstrates our resulting confusion matrix. Our model's ability to correctly identify positive cases (expert matching cases), measured in sensitivity/recall, was 56% (14/25). In

| |
|---|
| **SUMMARY:** Patient-1 has an *overall matching score of 0.5* with Protocol-1 |
| **NO EVIDENCE:** The program finds *NO* evidence that Patient-1 has a *cancer type* match with Protocol-1. |
| **EVIDENCE:** Patient-1 has a matching *ECOG status [1]* for Protocol-1 which requires ECOG status of 0-2. |
| **NO EVIDENCE:** The program finds *no evidence* Patient-1 has a mutation match with Protocol-1. |
| **EVIDENCE:** Patient-1 has a *measurable disease* and Protocol-1 requires one. |

**Fig 4. Example of our system output.**

**Table 1. Confusion matrix of patient-trial matching results.**

|  | Matching determined by our system | Non-Matching determined by our system | Total |
|---|---|---|---|
| Matching determined by physician | 14 | 11 | 25 |
| Non-Matching determined by physician | 5 | 42 | 47 |
| Total | 19 | 53 | 72 |

contrast, specificity, which reflects the ability to identify negative cases (expert non-matching cases) correctly, was 89.36% (42/47). Our model's overall correctness in both matching and non-matching cases assessed by accuracy was 77.78% (14+42/72), while precision in predicting matching cases (positive predictive value) was 73.68% (14/19).

## Incorrectly matched cases

**False negative cases.** For the eleven false negatives, upon assessment, they were found to be primarily from two protocols: protocols 2 and 4. Protocol 4 requires no specific cancer types. However, our program still detects some cancer types due to the misunderstanding of abbreviations and the context. For example, based on the original text: *"3) Liver function*: *Total bilirubin ≤ 1.5 × ULN (Subjects with Gilbert's Syndrome are allowed if total bilirubin ≤ 3 × ULN); AST and ALT ≤ 2.5 × ULN without liver metastases (≤ 5 × ULN if liver metastases are present); Albumin ≥ 2.5 g/dL."*, our program detects "ALT" as "Atypical Lipomatous Tumor," while in this context, ALT refers to "Alanine Transaminase." Our abbreviation detection and expansion function did not expand this abbreviation, as the protocol text does not mention the full meaning of this abbreviation. In addition, our program also detected "liver metastases" as a cancer type required by the protocol, while in this context, it refers to the condition of liver function requirement. The ambiguity of abbreviation and misunderstanding of the context contributed to nine false negatives by the system.

The other two false negatives are due to errors in genetic mutation matching. Our program missed the genetic mutation match between patient 4 and protocol 2 because the genetic mutation detected from the patient is "CHEK2, " a synonym of "CHK2" detected from the protocol. Currently, our program does not recognize synonyms of gene names. The other error in genetic mutation matching is due to the variations of expression reporting. In patient 5, the genetic mutation detected is "BRCA1 mut," while the corresponding genetic mutation detected in protocol 2 is "BRCA1".

**False positive cases.** For the five false positives, the system found evidence of no matching for all of them, even though the score is above 0.5 (Table 2). For example, patient 5 and protocol 3 pair have a matching score of 0.75, indicating 3 out of the 4 required criteria of protocol 3 are met by patient 5. However, the system did output "Patient 5 has ECOG status [2] which does not match Protocol 3 which requires ECOG status of 0–1". Similar reasons can also be found for other false positive matches (Table 2).

**Table 2. Reported evidence for non-match for false positive cases.**

| False positive matches | Reported evidence for non-match |
|---|---|
| Patient 5 and protocol 3 | ECOG no match reported |
| Patient 5 and protocol 6 | No cancer match reported |
| Patient 7 and protocol 2 | No mutation match reported |
| Patient 8 and protocol 2 | No mutation match reported |
| Patient 9 and protocol 5 | No mutation match reported |

## Discussion

### Misclassified cases

Among the misclassified cases, eleven false negatives and five false positives were identified. Regarding false negatives due to errors in genetic mutation matching in the case of using synonyms for certain genes, this can be addressed by incorporating a gene name thesaurus such as the Life Science Database Archive [26] that curates synonyms of gene names. While mismatches related to variations of expressions can be addressed by standardizing genetic mutation mentions in patients' records and trial protocols. As for cases where ambiguous use of abbreviations without clear definitions contributed to misclassification, improving structured reporting approaches in patient records and ensuring adequate expansion of abbreviations with the first mention could help limit such incorrect matches. For detected false positives, despite the system including evidence for non-matching in one of the four criteria for each patient, they were eventually deemed a match due to their matching scores being higher than 0.5. With a manual confirmation process, such evidence for non-matching related to one of the four criteria can be used to reject the matching decision by the system.

### Advantages of explainable AI

No decision-making system is perfect; however, a system that does not inform end-users what led to a particular decision (opaque or black-box systems) can maltreat some people in a population. Several measures are now being taken to reduce deployments of these systems and promote accountability, like European Union's General Data Protection Regulation [27] and U.S. Government's Algorithmic Accountability Act of 2019 [28]. Explanations are nothing but snippets of information that are deemed important by the system for a particular decision, thereby making a system more transparent. If a system is black-box then it might be prone to biased decisions that are distressing to certain populations [29]. Patient selection for any clinical trial is crucial for the well-being of the patient and their family, and there should be some accountability (both ethically and also lawfully in certain countries) for the decision-makers as to why somebody was rejected whereas someone else was selected, particularly when the selection is done using an automated system.

We employed an explainable AI approach when developing the system. This ensures that the system's output includes the explanation of the system decisions. Based on our detailed output evidence, any physician can understand why a particular score is assigned to a specific patient-protocol matching. The transparency of our system would give the information and trust that clinical trial research teams need while using such AI-aided systems. Also, the fact that our system does a preliminary check on four important criteria and provides concrete evidence for all of them would make the screening process much faster for physicians. Our reported performance metrics from the pilot study show promises of an AI-assisted approach for patient-trial matching in the future workflow of phase 1 oncology trials. It is worth mentioning that one practical advantage of our method is that it is not a black box system. Currently, many AI systems are black box systems that provide no explanation about the output [30]. This undermines the trustability and reliability of these systems, especially in a high-stake decision-making context like healthcare. Also, our model does not require patients' data to be in a tabular form; instead, unstructured text files are used for input, making the model more practical to suit the majority of patients' records format in clinical practice and research.

### Limitations and future directions

Several limitations should be noted in the current pilot study due to its initial stage. First, our model's performance may be compromised in cases with limited information pertaining to the

context of a specific criterion, as in the case of non-defining well-known abbreviations or using variable methods in reporting. In addition, specific phase 1 clinical trials require further detailed inclusion/exclusion criteria not covered by the current extraction and matching methods, such as previous treatment status as some trials limit eligibility to patients after certain therapies or a certain number of treatment lines. Lastly, the small sample size of processed patients' records and clinical trial protocols in this pilot project may not fully capture the complexity and variability encountered in real-world settings. A larger scale testing on more and real patient records is needed to demonstrate the use of the system in real-world settings.

This pilot study points out a few future directions that will further develop this system towards a practical AI-assisted patient-clinical-trial matching tool for phase 1 oncology trials:

- First, a larger sample of patient records and real patient records will be needed to validate the effectiveness of the system in complex and diverse real-world settings. The authors are in contact with a Clinical Research Data Warehouse to seek to include real patient records and a larger sample size in a future work.

- The four matching criteria implemented in this pilot study are selected based on the domain knowledge of the physician on the team. Although these criteria are deemed the most important for phase 1 oncology trials, they are not comprehensive. Future study will expand the eligibility criteria for phase 1 oncology trials to enhance the coverage of the system in screening patients.

- The analysis of the misclassified cases points out the need for continuous refinement of the NLP methods. The inclusion of ontology or thesaurus can help alleviate the problems of synonyms. The recent advancement in Large Language Models has shown impressive capabilities in understanding context [31]. Future work will explore the latest NLP techniques in addressing the drawbacks in the current system.

## Conclusion

In this study, we developed a novel method to match patients and phase 1 oncology clinical trials by combining several NLP tools and techniques. Our approach targets four criteria important for patient-clinical trial matching in the context of phase 1 oncology trials. The prototype system can produce a matching score and an explanation of evidence/no evidence of matching. This explainable AI-based approach is critical in the medical domain for trust and interpretability. To the best of our knowledge, no such system deploying an explainable AI-based approach currently exists in the public domain. This initial attempt achieved reasonable performance but was also presented with the limitation of misclassified cases. With further refinement and optimization, the eventual goal is to develop a system that will facilitate a decrease in study recruitment time, provider workload, and study cost while increasing the number of open trials, the precision of selection criteria, the amount of valuable data generated, and clinical trial success rates.

In conclusion, patient-clinical trial matching is a labor-intensive process in the current workflow of phase 1 oncology trials and limits patient recruitment for more efficient drug discoveries. This pilot study shows promises and feasibility of AI-assisted patient-clinical trial matching. Although the system's accuracy needs further improvement to support fully automated decision-making, the "evidence/no evidence" found by the system can already assist human decision-making and reduce the time required to assess patient-clinical trial matching for phase 1 oncology trials and increase the efficiency of new drug/treatment development. Future studies will further refine the NLP methods, incorporate more inclusion/exclusion

criteria that are important for phase 1 oncology trials, and increase the sample size for more robust testing. We envision a human-AI collaborative approach to patient-clinical trial matching in future phase 1 oncology trials workflow.

## Supporting information

**S1 Checklist. Human participants research checklist.**
(DOCX)

## Author Contributions

**Conceptualization:** Abdul Rafeh Naqash, Kun Lu.

**Formal analysis:** Satanu Ghosh, Kun Lu.

**Funding acquisition:** Chongle Pan, Abdul Rafeh Naqash, Kun Lu.

**Investigation:** Chongle Pan, Abdul Rafeh Naqash, Kun Lu.

**Methodology:** Satanu Ghosh, Kun Lu.

**Project administration:** Kun Lu.

**Software:** Satanu Ghosh, Arjun Ganesan, Kun Lu.

**Supervision:** Kun Lu.

**Validation:** Kun Lu.

**Writing – original draft:** Satanu Ghosh, Arjun Ganesan, Abdul Rafeh Naqash, Kun Lu.

**Writing – review & editing:** Satanu Ghosh, Hassan Mohammed Abushukair, Chongle Pan, Kun Lu.

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
