## [Decision Letter · Decision Letter 0]

15 Jul 2024

PONE-D-24-02458Harnessing explainable Artificial Intelligence for Patient-to-Clinical-Trial matching: A proof-of-concept pilot study using Phase I Oncology TrialsPLOS ONE

Dear Dr. Lu,

Thank you for submitting your manuscript to PLOS ONE. After careful consideration, we feel that it has merit but does not fully meet PLOS ONE’s publication criteria as it currently stands. Therefore, we invite you to submit a revised version of the manuscript that addresses the points raised during the review process.

Please carefully check the Reviewers’ comments and improve the manuscript. Reviews provide details into areas that require improvement.

We look forward to receiving your revised manuscript.

Kind regards,

Agnieszka Konys, Ph.D.

Academic Editor

PLOS ONE

“This work is supported by an internal seed fund at The University of Oklahoma.”

4. In the online submission form, you indicated that [Patient records cannot be shared publicly due to the HIPPA requirement. The output of the system can be shared through Google drive].

Reviewers' comments:

Reviewer's Responses to Questions

**Comments to the Author**

1. Is the manuscript technically sound, and do the data support the conclusions?

Reviewer #1: Yes

Reviewer #2: Partly

Reviewer #3: Partly

2. Has the statistical analysis been performed appropriately and rigorously? 

Reviewer #1: No

Reviewer #2: No

Reviewer #3: No

3. Have the authors made all data underlying the findings in their manuscript fully available?

Reviewer #1: No

Reviewer #2: No

Reviewer #3: No

4. Is the manuscript presented in an intelligible fashion and written in standard English?

Reviewer #1: Yes

Reviewer #2: Yes

Reviewer #3: No

5. Review Comments to the Author

Reviewer #1: Strengths:

Innovative Approach: The study introduces an innovative use of explainable AI to the domain of clinical trial matching, focusing on phase I oncology trials. This approach is significant as it aims to increase the efficiency of patient recruitment without compromising the quality of patient-trial matching.

Use of NLP Techniques: The application of modern NLP techniques to process unstructured patient records and clinical trial protocols represents a substantial advancement over traditional manual matching methods. This could potentially streamline the matching process, making it faster and more accurate.

Explainability and Transparency: One of the key contributions of this study is the emphasis on explainability in the AI matching process. Providing a summary matching score along with explanations for the evidence contributes to the transparency and trustworthiness of the AI system, which is crucial in a clinical setting.

Pilot Study Design: The proof-of-concept nature of the study, demonstrated through a pilot system, showcases the feasibility of the proposed approach. The detailed analysis of the system's performance, including precision, sensitivity/recall, accuracy, and specificity metrics, provides a solid foundation for further development and refinement.

Limitations:

Sample Size and Data Limitations: The study is based on a relatively small dataset of synthesized dummy patient records and clinical trial protocols, which may not fully capture the complexity and variability encountered in real-world settings. Future work would benefit from larger and more diverse datasets to validate and improve the system's performance.

Misclassified Cases Analysis: The manuscript discusses the instances of misclassification, attributing errors to ambiguity in abbreviations, misunderstanding of context, and variations in expression. These insights are valuable, but they also highlight the need for continuous refinement of the NLP models and preprocessing steps to handle such complexities more effectively.

Scope for Expanding Eligibility Criteria: The study currently focuses on four main criteria for matching. However, clinical trial eligibility often involves a wider range of criteria. Expanding the system to consider additional criteria could enhance its applicability and accuracy.

Future Directions:

The manuscript outlines several promising areas for future research, including the integration of gene name thesauri to address genetic mutation matching errors and the exploration of structured reporting approaches to reduce ambiguity. Furthermore, extending the prototype system to include more detailed inclusion/exclusion criteria and testing it on a larger scale are essential steps towards realizing a practical AI-assisted patient-clinical trial matching tool.

Reviewer #2: The research is emerging with the application of explanation AI. However, more detailed analysis how explainable AI is used in the proposed work is suggested.

2.The authors should consider some explanation without explainable AI and discuss the improvement achieved.

3. How, the proposed method is trustworthy, transparent and secured as mentioned by the authors, to be elaborated with more detail .

4. Some comparison with other approaches to be discussed with their pitfalls, even though the authors states no research is carried out is novel.

5. How the proposed preprocessing and feature extraction process carried out and what were the outputs and which AI technique is used, please present it in a flow chart and if possible, write down the pseudocode for them.

Reviewer #3: This work propose using an AI system, that utilizes modern NLP methods to match patient records with clinical trial protocols based on four criteria: cancer type, performance status, genetic mutation, and measurable disease. It provides a matching score and evidence-based explanations. While the idea of the model is great and may help in finding th epropoer clinical triials. I have major concern about the depth of the information and number of samples as the following:

- More in depth data are required to build the relationships between gene mutations+ clinical data with the clinical trial. For example, full variant information included the position the reference and the change, and more in depth about coverage.

- A lack of number of samples in this study

- A lack of proper measurements (evaluations) out of this study.

Again, this is a good idea but still immature for publication.

6. PLOS authors have the option to publish the peer review history of their article (what does this mean?). If published, this will include your full peer review and any attached files.

Reviewer #1: No

Reviewer #2: No

Reviewer #3: **Yes: **Abedalrhman Alkhateeb

---

## [Author Response · Author response to Decision Letter 0]

6 Sep 2024

August 30th, 2024

Dear editor,

 We thank you very much for your work on processing this submission. We appreciate reviewers’ feedback and comments. Below please find our responses to reviewers. Along with this letter, please also find a marked-up copy of the manuscript that highlights all changes labelled “Revised Manuscript with Track Changes,” and an unmarked version of the revised paper labelled “Manuscript.” We have also addressed the journal requirements in this revision. Please feel free to reach out if you have any questions.

Sincerely,

Kun Lu on behalf of coauthors

Response: We have revised the manuscript to meet PLOS ONE style requirements. 

Response: The authors of this publications are seeking a patent application from this work. Sharing code publicly at this point would constitute a public disclosure of the work and conflicts with the patent application. We have added information on how to request access to the code in the Data Availability Statement in the revised manuscript.

“This work is supported by an internal seed fund at The University of Oklahoma.”

Response: The funders had no role in the study. We have included the amended Role of Funder statement in the cover letter.

4. In the online submission form, you indicated that [Patient records cannot be shared publicly due to the HIPPA requirement. The output of the system can be shared through Google drive].

Response: These data can't be shared. These data contain potential patient identifiers and genomic information that can't be shared openly in the journal. Select deidentified data can be potentially shared with readers upon specific requests to the corresponding authors. This is imposed by the Institutional Review Board based on HIPAA.

Reviewer #1: 

Strengths:

Innovative Approach: The study introduces an innovative use of explainable AI to the domain of clinical trial matching, focusing on phase I oncology trials. This approach is significant as it aims to increase the efficiency of patient recruitment without compromising the quality of patient-trial matching.

Use of NLP Techniques: The application of modern NLP techniques to process unstructured patient records and clinical trial protocols represents a substantial advancement over traditional manual matching methods. This could potentially streamline the matching process, making it faster and more accurate.

Explainability and Transparency: One of the key contributions of this study is the emphasis on explainability in the AI matching process. Providing a summary matching score along with explanations for the evidence contributes to the transparency and trustworthiness of the AI system, which is crucial in a clinical setting.

Pilot Study Design: The proof-of-concept nature of the study, demonstrated through a pilot system, showcases the feasibility of the proposed approach. The detailed analysis of the system's performance, including precision, sensitivity/recall, accuracy, and specificity metrics, provides a solid foundation for further development and refinement.

Limitations:

Sample Size and Data Limitations: The study is based on a relatively small dataset of synthesized dummy patient records and clinical trial protocols, which may not fully capture the complexity and variability encountered in real-world settings. Future work would benefit from larger and more diverse datasets to validate and improve the system's performance.

Response: Thanks for the comment. We agree that the small dataset is a limitation of this pilot study. We have acknowledged this as a limitation of the study in the manuscript and plan to seek resources for a larger follow-up study involving more and real patient records. This pilot study only serves as a proof-of-concept purpose to convince a larger follow-up study is feasible. However, we think this work is still valuable as there has not been any explainable-AI-based systems currently deployed for patient-clinical-trials matching for phase 1 ontology trials and the initial functions of the system can already be helpful in assisting patient-clinical-trial matching. This pilot study also points out future directions of the development. We have reorganized the content of the manuscript and added a dedicated section “Limitations and Future Directions” in the Discussion section to discuss the limitations and future directions of this study.

Misclassified Cases Analysis: The manuscript discusses the instances of misclassification, attributing errors to ambiguity in abbreviations, misunderstanding of context, and variations in expression. These insights are valuable, but they also highlight the need for continuous refinement of the NLP models and preprocessing steps to handle such complexities more effectively.

Response: We agree with the reviewer that continuous refinement of NLP methods is needed to improve the performance of the system. This is a future direction we plan to explore. We have added this to the “Limitations and Future Directions” section to emphasize this need.

Scope for Expanding Eligibility Criteria: The study currently focuses on four main criteria for matching. However, clinical trial eligibility often involves a wider range of criteria. Expanding the system to consider additional criteria could enhance its applicability and accuracy.

Response: Thanks for the comment. We agree that phase 1 oncology trials involve many more matching criteria than what we have considered (e.g. previous treatment status of the patients as some trials limit eligibility to patients after certain therapies or a certain number of treatment lines). The selection of the four main criteria in this pilot study is based on the experience of the physician on the team to include the most important ones. We do plan to consider additional matching criteria as we further develop this line of research. 

Future Directions:

The manuscript outlines several promising areas for future research, including the integration of gene name thesauri to address genetic mutation matching errors and the exploration of structured reporting approaches to reduce ambiguity. Furthermore, extending the prototype system to include more detailed inclusion/exclusion criteria and testing it on a larger scale are essential steps towards realizing a practical AI-assisted patient-clinical trial matching tool.

Response: Thanks! Yes, we agree. We have outlined these in the new section on “Limitations and Future Directions” in Discussion.

Reviewer #2: 

The research is emerging with the application of explanation AI. However, more detailed analysis how explainable AI is used in the proposed work is suggested.

2.The authors should consider some explanation without explainable AI and discuss the improvement achieved.

Response: Thanks for the comment. We realize how this part was lacking in our paper and therefore we have added a paragraph (3rd paragraph) in the “Introduction” section and the second part of “Discussion” section about implications of using explainable v/s. black-box model in healthcare or similar domains.

3. How, the proposed method is trustworthy, transparent and secured as mentioned by the authors, to be elaborated with more detail .

Response: Thanks! The idea of a transparent system is to provide evidence for predictions. Our method is trustworthy and transparent because it covers two important aspects: 1) it provides not with binary prediction but rather a score implying the strength of matching, and 2) evidence that contributes toward the score is also provided to the end-users. So, ultimately it is up to the physician to make the final call, based on the score and the evidence cumulatively. For example, if a protocol serves a very specific cancer type and the patient does not have that type of cancer then the patient should not be selected even if all the three other criteria matched. But we leave it to the oncologists and only provide information to take the final decision. Our approach reduces the cognitive load from a physician by providing important information but does not take a final decision.

4. Some comparison with other approaches to be discussed with their pitfalls, even though the authors states no research is carried out is novel.

Response: Thank you for the comment. As we mentioned that no previous method exists for Phase1 Clinical Trial for Oncology, so we could not compare it directly to any other method. However, we added some relevant literature in the “Introduction” and discuss their pitfalls and why this study is uniquely poised.

5. How the proposed preprocessing and feature extraction process carried out and what were the outputs and which AI technique is used, please present it in a flow chart and if possible, write down the pseudocode for them.

Response: Thanks! We have added a flow-chart (Figure 1) in Section 3. Also, we want to point towards: Figure 2: describes the tools and normalizations that were performed in the pre-processing step, Figure 3: show the output of the pre-processing step, and Figure 4: contains an example output of the system. To explain the nature of AI technique we have added the following text: “… an unsupervised prototype NLP system that follows a mixed approach combining tools, regular expressions and expert curated rules …” in the description of our work (can be found in the last paragraph of “Introduction”).

Reviewer #3: 

This work proposes using an AI system, that utilizes modern NLP methods to match patient records with clinical trial protocols based on four criteria: cancer type, performance status, genetic mutation, and measurable disease. It provides a matching score and evidence-based explanations. While the idea of the model is great and may help in finding the proper clinical trials. I have major concern about the depth of the information and number of samples as the following:

- More in depth data are required to build the relationships between gene mutations+ clinical data with the clinical trial. For example, full variant information included the position the reference and the change, and more in depth about coverage.

Response: We thank the reviewer for the insightful comment. However, while we agree that more in-depth genomic data would be important for therapy response or sensitivity, that is not the focus of this current body of work and is not relevant to this manuscript at this point. We are focused on NLP and textual information in this study.

- A lack of number of samples in this study

Response: Thanks for the comment. We acknowledge the small sample size is a limitation of this pilot study. We are using this pilot study as a proof-of-concept to seek resources for a larger follow-up study involving more and real patient records. We have added a subsection in the Discussion section on “Limitations and Future Directions.” However, we believe this pilot study is still valuable as there has not been any explainable-AI-based systems currently deployed for patient-clinical-trials matching for phase 1 ontology trials. In addition, some functions of the system can already be helpful in assisting patient-clinical-trial matching.

- A lack of proper measurements (evaluations) out of this study.

Response: Thanks for the comment. Our system is developed to assist matching patients with phase 1 oncology trials. The matching decisions can be considered as a classification problem with the outcomes to be either match or non-match. We therefore evaluate the system accordingly using the classic confusion matrix, and then report metrics such as accuracy, precision, recall, specificity etc. The results provide insights into where the system made correct decisions (the ones match those from a physician) and where the system produced false positives or false negatives. This is a common practice in evaluating classification problems and should have provided sufficient information on the performance of the system. We kindly request the reviewer to clarify if there is some other information on evaluation is needed.

---

## [Editor Report · Decision Letter 1]

20 Sep 2024

Harnessing explainable Artificial Intelligence for Patient-to-Clinical-Trial matching: A proof-of-concept pilot study using Phase I Oncology Trials

PONE-D-24-02458R1

Dear Dr. Lu,

We’re pleased to inform you that your manuscript has been judged scientifically suitable for publication and will be formally accepted for publication once it meets all outstanding technical requirements.

Kind regards,

Agnieszka Konys, Ph.D.

Academic Editor

PLOS ONE
---

## [Editor Report · Acceptance letter]

11 Oct 2024

PONE-D-24-02458R1 

PLOS ONE

Dear Dr. Lu, 

I'm pleased to inform you that your manuscript has been deemed suitable for publication in PLOS ONE. Congratulations! Your manuscript is now being handed over to our production team.

Kind regards, 

on behalf of

Dr. Agnieszka Konys 

Academic Editor

PLOS ONE